# Children and adolescents: Respiratory infection and long-term effects longitudinal study (CARE Study): Study protocol

Mark McMillan [1,2*], Rebecca Beazley[3], Nan Vasilunas[4], Thomas R. Sullivan[5,6],
Tess Edmond[7], Ailish Battersby[7], Philip N. Britton [8,9], Brendan McMullan[10,11],
Jon Jureidini[12], Sarah Del Fante[3], Helen S. Marshall[1,2]

**1** University Department of Paediatrics, Women's and Children's Hospital Network, North Adelaide, South Australia, Australia, **2** Adelaide Medical School and Robinson Research Institute, University of Adelaide, North Adelaide, South Australia, Australia, **3** Communicable Disease Control Branch, South Australia Health, Adelaide, South Australia, Australia, **4** Infectious Disease Department, Women's and Children's Hospital Network, North Adelaide, South Australia, Australia, **5** Women and Kids Theme, South Australian Health and Medical Research Institute, Adelaide, South Australia, Australia, **6** School of Public Health, The University of Adelaide, Adelaide, South Australia, Australia, **7** Department of General Medicine, Women's and Children's Hospital Network, North Adelaide, South Australia, Australia, **8** Department of Infectious Diseases and Microbiology & Child and Adolescent Health, Children's Hospital Westmead, Westmead, Australia, **9** Sydney Medical School, University of Sydney, Camperdown, New South Wales, Australia, **10** Department of Infectious Diseases, Sydney Children's Hospital, Randwick, New South Wales, Australia, **11** School of Clinical Medicine, Faculty of Medicine and Health, University of New South Wales, Kensington, New South Wales, Australia, **12** Critical and Ethical Mental Health (CEMH), Robinson Research Institute, University of Adelaide, Adelaide, South Australia, Australia

* mark.mcmillan@adelaide.edu.au

## Abstract

### Background

The effects of SARS-Cov-2 infection can extend beyond the acute phase of the illness, often described as Long COVID, post-COVID condition (PCC) or Post-acute sequelae of COVID (PASC). Post-acute sequelae (PAS) are also likely to be a problem for a small proportion of children and adolescents following influenza infection. However, there is no comprehensive ongoing data collection in Australian children and adolescents, and global data on both PCC during the SARS-Cov-2 Omicron variant period and PAS following influenza is limited.

### Aim

This study aims to determine the cumulative incidence of PCC in Australian children and adolescents five years after the start of the COVID-19 pandemic. Secondary aims include identifying the cumulative incidence of PAS in children and adolescents following influenza infection.

**Data availability statement:** No datasets were generated or analysed during the current study. All relevant data from this study will be made available upon study completion.

**Funding:** Dr. Mark McMillan (MM) is supported by a Women's & Children's Hospital Foundation Postdoctoral Fellowship. Funder Name: Women's & Children's Hospital Foundation Website: https://wchfoundation.org.au Grant Number: Not applicable The funder had no role in study design, data collection and analysis, decision to publish, or preparation of the manuscript.

**Competing interests:** The authors have declared that no competing interests exist.

## Methods

This longitudinal cohort study will recruit children and adolescents aged 0–18 years in South Australia who tested positive for SARS-Cov-2 or influenza in the previous 2 months. Following consent, participants will complete an online baseline survey and then at 3, 6, and 12 months post-infection. The survey has been adapted from the International Severe Acute Respiratory and Emerging Infection Consortium (ISARIC) Paediatric COVID-19 follow-up survey. The survey includes validated assessment tools such as the Pediatric Quality of Life Inventory (PedsQL), Multidimensional Fatigue Scale, and the Malmö Postural Orthostatic Tachycardia Syndrome (POTS) Score questionnaire. PCC following COVID-19 and PAS following influenza infection will be identified according to an adapted World Health Organization definition of PCC in children and adolescents.

## Discussion

This study addresses gaps in understanding PCC and PAS following influenza in children and adolescents during Omicron circulation. Whilst it is no longer feasible to prospectively compare post-acute sequelae in children and adolescents who have never had COVID-19, this design allows a comparison with another common viral infection, influenza, informing clinical management of children post-infection.

## Introduction

The effects of SARS-Cov-2 infection can extend beyond the acute phase of the illness, with complications persisting across multiple body systems in the months and, in some cases, years following infection. These ongoing conditions are described as post-acute sequelae resulting from COVID-19 infections, post-COVID-19 condition (PCC) or long COVID [1]. For consistency, the term PCC will be used throughout this document. The incidence, symptoms, and duration of PCC differ between children and adults [2]. Children are also at risk of developing the rare post-infectious inflammatory condition, multisystem inflammatory syndrome or paediatric inflammatory, multisystem syndrome temporally associated with SARS-Cov-2, which occurs in approximately 1 in every 2,500 cases. While this condition is predominantly seen in children, adult cases have been reported [3,4].

Recognizing a need for a standardized definition, a panel of 27 experts convened in September 2022 to define PCC in children and adolescents, which was ratified by the World Health Organization (WHO) [5]. The criteria include 1) a history of confirmed or probable SARS-CoV-2 infection, 2) symptoms lasting at least 2 months, initially occurring within 3 months of acute infection, 3) symptoms generally have an impact on everyday functioning, 4) symptoms may be new onset following initial recovery from an acute COVID-19 episode or persist from the initial illness, and may fluctuate or relapse over time, and 5) workup may reveal additional diagnoses to PCC [5,6].

## Current evidence

Multiple systematic reviews of the long-term effects of SARS-CoV-2 infection on children and adolescents have been performed, [7–14] with some focusing on specific areas such as mental health outcomes, [15] respiratory outcomes, [16,17] cardiovascular outcomes, [18] new-onset type 1 diabetes, [19] and risk factors for PCC [20]. Only a small proportion (approximately 10%) of studies in the systematic reviews reported PCC during the current Omicron period, which was first identified on the 25th of November 2021 [14,21]. One of the more recent reviews included 55 studies (1,139,299 participants) published before November 2022 and defined PCC symptoms as lasting at least 12 weeks post-infection [11]. The primary analysis was restricted to controlled studies, as the authors highlighted the limitations of uncontrolled studies due to the nonspecific nature of some symptoms, such as headache and fatigue. They also noted that it is now difficult to conduct studies with non-COVID-19 controls due to the high proportion of the population that has had COVID-19 at least once. Their meta-analysis only included 11 controlled studies (n = 292,978) and reported significantly higher pooled estimates for altered/ loss of smell or taste (risk difference [RD] 4%), dyspnoea (RD 3%), fatigue (RD 4%), and myalgia (RD 1%) 12 weeks post-COVID-19 infection compared to control groups [11].

Despite a reasonable body of research into the incidence of PCC in children, most data is from Europe, and during the pre-Omicron period, only two studies in children have been conducted in Australia. The first is a study of 171 children from Melbourne that found that the most common PCC symptoms were mild post-viral cough (6/151 [4%]), fatigue (3/151 [2%]) and post-viral cough and fatigue (1/151 [1%]) [22]. The second study was conducted at the Sydney Children's Hospitals Network (SCHN) during the Delta outbreak. The study contacted 9,765 eligible children (aged 0–18 years) and their families. Out of these, 1,731 responded. From the responses, 203 (11.7%) reported continued symptoms and/or functional impairment, with 169 individuals receiving follow-up from a clinician. From the final responder cohort, 1.2% (95% CI 0.8 to 1.8) had persistent symptoms impacting daily function comparable with the UK consensus definition of PCC [23].

## Influenza as a comparator for post-viral conditions

Unlike COVID-19, there is no accepted definition for post-acute sequelae of influenza. However, adults can experience central nervous system (CNS) syndromes, such as Guillain-Barre Syndrome (GBS), and a worsening of underlying conditions like ischemic heart disease and cerebrovascular disease following influenza infection. These health issues can manifest weeks to months after recovery [24].

Other studies have made comparisons between post-acute sequelae following COVID-19 and influenza in adults. A 6-month retrospective cohort study utilizing electronic medical records of 236,379 COVID-19 patients and 105,579 diagnosed with influenza identified higher rates of anxiety and mood disorders, insomnia, and dementia post-COVID-19 than post-influenza [25]. Another, which analyzed electronic medical records from American veterans (88% male), reported increased rates of sequelae in multiple body systems following COVID-19 compared to influenza, including neurological and neurocognitive disorders, mental health disorders, cardiovascular disorders, gastrointestinal disorders, coagulation disorders, and other disorders, including malaise and fatigue [26]. A phone survey of Queensland adults revealed minimal differences in PCC and PAS following influenza 12 weeks after an Omicron outbreak [27]. Unlike children, this cohort was a highly vaccinated population.

With near-universal COVID-19 exposure, comparing PCC against non-COVID-19 controls is now extremely difficult. Influenza serves as a practical alternative to assess post-infectious sequelae in children, providing a comparator for distinguishing PCC from non-COVID post-acute sequelae. An additional benefit of including an influenza control group is establishing the long-term impacts of influenza on children and adolescents, which remains poorly described.

## Reinfection and remaining evidence gaps

Despite COVID-19 circulation now in its 5th year, there remains an ongoing risk for children and adolescents regarding PCC. The United Kingdom's (UK) CLoCk Study, which carried out a longitudinal cohort study on 11–17-year-olds with

both negative and positive COVID-19 tests, found that young people with COVID-19 reinfection had more symptoms and a higher prevalence of symptoms than those following their first infection [1]. Another survey led by the Office for National Statistics in the UK of over 500,000 participants from the UK showed no reduction in risk of developing PCC in <16 year-olds for those experiencing a second infection, compared to the first infection [28].

### Study aims

This study aims to identify the cumulative incidence of PCC and PAS following influenza in Australian children and adolescents five or more years after the start of the COVID-19 pandemic.

## Methods

This study is designed as a longitudinal case-cohort study. The study will recruit participants with COVID-19 or influenza, and each participant will be enrolled for 12 months. They will receive an online survey at baseline, then 3, 6, and 12 months post-infection.

### Objectives

#### Primary objective.

1. Estimate the cumulative incidence of PCC in children and adolescents with COVID-19 infection at 3 months, and the proportion with ongoing symptoms at 6 and 12 months post-COVID-19 infection.

#### Secondary objectives.

1) Estimate the cumulative incidence of PAS in children and adolescents with influenza infection at 3 months, and the proportion with ongoing symptoms at 6 and 12 months.

2) Estimate the relative risk of developing PCC following COVID-19 compared to PAS following influenza at 3 months post-infection.

3) Develop a prediction model for the development of PCC in children and adolescents with COVID-19.

4) Estimate the difference in the quality of life of children and adolescents (aged 2–18) at 3, 6, and 12 months post-infection in the following groups:

   i) PCC compared to those without PCC in COVID-19-positive cases.
   ii) PAS compared to those without PAS in influenza-positive cases.
   iii) PCC compared to those with PAS

5) Estimate the difference in fatigue of children and adolescents (aged 2–18) at 3, 6, and 12-months post-infection in the following groups:

   i) PCC compared to those without PCC in COVID-19-positive cases.
   ii) PAS compared to those without PAS in influenza-positive cases.
   iii) PCC compared to those with PAS

6) Estimate the difference in school/childcare absenteeism at 3, 6, and 12 months following infection in the following groups:

   i) PCC compared to those without PCC in COVID-19-positive cases.
   ii) PAS compared to those without PAS in influenza-positive cases.
   iii) PCC compared to those with PAS

7) Estimate the difference in influenza vaccine uptake in children and adolescents with PAS following influenza compared to vaccination uptake in those without PAS following influenza.

8) Estimate the difference in the cumulative incidence of PCC in those with COVID-19 re-infections compared to those with first infection at 3 months, and the proportion with ongoing symptoms at 6, and 12 months post-infection.

9) Describe the duration and type of PCC in children and adolescents with COVID-19 infection.

10) Describe the duration and clinical features of PAS in children and adolescents following influenza infection.

## Participants

Children and adolescents aged 0–18 who have tested SARS-CoV-2 positive (PCR or Rapid Antigen Tests) or influenza (PCR or Rapid Antigen Tests) in the previous 2 months are eligible for enrolment in South Australia, which has a population of approximately 1.8 million people, and 374,000 children and young people under the age of 18 [29].

## Exclusion criteria

1. Inability to give informed parental consent

2. No mobile phone number available

3. Children and adolescents who are under the care of the statewide palliative care service

## Recruitment

Recruitment includes children and young adults aged 0–18 with COVID-19 and influenza infections from South Australia-wide notifications. These notifications include children and young adults aged 0–18 who have not presented to a hospital setting. Under the South Australian Public Health Act 2011, medical practitioners and diagnostic laboratories are required to notify SA Health of cases with SARS-Cov-2 or influenza infections [30]. An SMS invitation is sent to those eligible that contains a link to the participant information sheet and consent form. The study commenced recruitment on May 6, 2024, with each participant enrolled for a duration of 12 months, and recruitment to finish on June 30, 2026. Each participant receives an AUS $10 supermarket voucher for completing each survey, compensating them for their time ($40 total for 4 surveys).

## Consent

Written informed consent will be obtained from parents or guardians for all participants. For participants aged 12 years or older who wish to complete the surveys themselves, written assent will also be obtained. All consent and assent responses are recorded electronically.

## Outcome measures

**PCC cases.** PCC cases will be identified using a modified version of the World Health Organization's definition of PCC in children and adolescents, requiring confirmed infection, symptoms that begin within 3 months and persist for at least 2 months (S1 Appendix). Cases will be further characterized by the presence or absence of an impact on everyday functioning [5]. This assessment will use survey responses and electronic medical record reviews where applicable. Survey results from participants who may meet the criteria for PCC or PAS will be reviewed by a multi-specialty panel, blinded to their baseline COVID-19 or influenza status, for classification.

**PAS following influenza.** PAS cases following influenza will be assessed using a modified version of the World Health Organization's definition of PCC in children and adolescents as described above.

## Participant data and study instruments

Following consent, survey data will be collected from enrolled participants using an online RedCap baseline question-naire, which will be completed at recruitment. This will be followed by questionnaires at 3, 6, and 12 months post-infection. Study data, including consent, are collected and managed using a secure REDCap electronic data capture tool hosted at Central Adelaide Local Health Network, SA Health [31]. Questionnaires are available for parents/carers or participants to answer, where applicable. The survey is based on the International Severe Acute Respiratory and Emerging Infection Consortium (ISARIC) COVID-19 Paediatric follow-up protocol and questionnaire [32]. The impact on everyday functioning is assessed using the Pediatric Quality of Life Inventory (PedsQL) short form. The 15 items in the PedsQL comprise four Generic Core Scales (for ages 2–18): Physical Functioning, Emotional Functioning, Social Functioning, and School Functioning [33]. Fatigue is assessed using the PedsQL Multidimensional Fatigue Scale (for ages 2–18), which is composed of 18 items across three dimensions: General Fatigue, Sleep/Rest Fatigue, and Cognitive Fatigue [34]. The Malmö Postural Orthostatic Tachycardia Syndrome (POTS) score (MAPS) questions are also included [35]. An adapted SPIRIT schedule is outlined in (Fig 1) [36]. Outliers will be queried during data cleaning. The final identifiable dataset will be accessible only to the Principal Investigator.

## Statistical analysis

Analyses will follow a pre-specified statistical analysis plan. The cumulative incidence of PCC following COVID-19 (primary objective) and PAS following influenza will be estimated at 3 months post-infection, with 95% confidence intervals. The proportion of individuals with ongoing symptoms will be reported descriptively (n, %) at 6 and 12 months. The relative risk of developing PCC following COVID-19 versus developing PAS following influenza (objective 2) will be estimated using a log-binomial model. A multivariable logistic model will be developed for predicting PCC at 3 months (objective 3), incorporating pre-specified predictors such as age (treated as continuous), sex, pre-existing conditions, and socioeconomic status. Model performance will be evaluated using the area under the receiver operating characteristic curve, with internal validation by boot-strap resampling in line with TRIPOD recommendations [37]. The relatively small expected number of PCC events may limit the number of predictors that can be incorporated into the final model and hence its predictive accuracy. Linear and negative binomial mixed-effects models will assess differences between comparator groups over time in continuous (PedsQL quality of life and fatigue scores, objectives 4 and 5) and count (days absent, objective 6) outcomes, respectively. Log binomial models will compare the risk of developing PAS according to influenza vaccine uptake (objective 7) and the risk of PCC by COVID-19 reinfection (objective 8). The duration and clinical features of PCC and PAS will be described using summary statistics, with summaries at 3, 6 and 12 months. Descriptive analyses will be based on complete case data.

## Handling of follow-up and missing data

Outcomes will be summarised at the planned 3, 6 and 12-month assessments. Participants contribute data to each anal-ysis for which follow-up information is available. For the primary objective, participants with PCC at 3 months who dis-continue or miss assessments at 6 or 12 months will be treated as missing for whether symptoms were ongoing at these time-points. Multiple imputation may be considered for the prediction model, depending on the amount and reasons for missing data at 3 months when determining PCC.

## Sample size

There are expected to be approximately 1,222 COVID-19 cases, with 2,800 influenza cases recruited over the 24 months of the study, assuming a survey response rate of 15%. If the PCC incidence is between 1% to 6%, a sample size of 1,222

| TIMEPOINT | Exposure -t₁ | Enrolment 0 | Post-exposure t₁ Baseline | t₂ 3 months | t₃ 6 months | T₄ 12 months | Close-out |
|---|---|---|---|---|---|---|---|
| **EXPOSURE CLASSIFICATION:** | | | | | | | |
| Positive test for COVID-19 or influenza | ✔ | | | | | | |
| Eligibility screen | ✔ | | | | | | |
| **ENROLMENT:** | | | | | | | |
| Informed parental consent | | ✔ | | | | | |
| Informed assent (if 12 years of age or older) | | ✔ | | | | | |
| Eligibility screen | | ✔ | | | | | |
| **INTERVENTIONS:** | | | | | | | |
| No intervention – observational study | | | | | | | |
| **ASSESSMENTS:** | | | | | | | |
| **Demographics** | | | | | | | |
| Date of birth | | | ✔ | | | | |
| Postcode | | | ✔ | | | | |
| Sex | | | ✔ | | | | |
| **Clinical Information** | | | | | | | |
| Concomitant infection | | | ✔ | | | | |
| History of COVID-19 infection and influenza | | | ✔ | ✔ | ✔ | ✔ | |
| Pre-existing medical conditions | | | ✔ | | | | |
| Symptoms of acute infection (first 14 days) | | | ✔ | | | | |
| Vaccination history (from register) | | | | | | ✔ | |
| **Hospitalisation information (if applicable)** | | | | | | | |
| Length of stay, treatments, ICU | | | ✔ | ✔ | ✔ | ✔ | |
| **Ongoing Health** | | | | | | | |
| Physical and mental health symptoms (ISARIC list) | | | | ✔ | ✔ | ✔ | |
| Malmö POTS score (MAPS) | | | | ✔ | ✔ | ✔ | |
| New medically diagnosed conditions | | | | ✔ | ✔ | ✔ | |
| Days absent from school/work | | | | ✔ | ✔ | ✔ | |
| **Quality of Life and Fatigue** | | | | | | | |
| Pediatric quality of life inventory (PedsQL). | | | | ✔ | ✔ | ✔ | |
| PedsQL Multidimensional Fatigue Scale | | | | ✔ | ✔ | ✔ | |
| **Assessment of PCC and PAS (panel)** | | | | | | | ✔ |
| **Final data collection & study close-out** | | | | | | | ✔ |

ICU = Intensive Care Unit, HDU = High Dependency Unit, ISARIC = International Severe Acute Respiratory and Emerging Infection Consortium, POTS = Postural Orthostatic Tachycardia Syndrome.

**Fig 1. SPIRIT schedule of enrolment.**

COVID cases allows for the incidence to be estimated with a precision ranging from 0.62% to 1.47%, with 20% loss to follow-up over 12 months (with precision defined as the width of a 95% CI around the estimated incidence) Table 1.

## Ethical approval, study registration, and funding

This study was approved by the Women's and Children's Health Network (WCHN) Human Research Ethics Committee, 2023/HRE00283. The study is registered at the Australian New Zealand Clinical Trials Registry, ACTRN12624001361594 [38]. Any protocol changes will be submitted for review by the WCHN HREC and updated on the trial registry.

## Consumer engagement

The study has two consumer representatives with lived experience of caring for a child with PCC. The consumers are involved in reviewing all study material, including the protocol, PI sheet, consent form, questionnaires, and

**Table 1. Precision estimates based on either the proportion of long COVID in the population (between 1-6%) and the sample size (between 550-2000).**

| | Proportion with PCC | | |
| | 1% | 3% | 6% |
| --- | --- | --- | --- |
| Sample size | Margin of error for incidence estimate* | | |
| 550 | 0.83% | 1.42% | 1.98% |
| 1,000 | 0.62% | 1.06% | 1.47% |
| 2,000 | 0.43% | 0.75% | 1.04% |

*Assume 360,000 population at risk (children under 18 in South Australia)

communications with potential participants and participants. The consumers will also be involved in interpreting results, planning communication strategies, and disseminating the results.

### Safety considerations

Standard care for individuals under 18 years of age who experience ongoing sequelae from a COVID-19 infection or non-COVID respiratory infections primarily involves assessment through general practice or hospital outpatient services. Children who report persistent severe symptoms and have not yet undergone a medical assessment will be contacted for a nurse-led telehealth appointment to determine if a referral to their General Practitioner is required.

## Discussion

There is no comprehensive approach to collecting data on PCC in Australia, which is required to inform management of the condition and the implementation of interventional studies [23]. While there is currently no definitive treatment, COVID-19 vaccination has demonstrated some protection in adults [39] and children aged 5–17, with an effectiveness of 41.7% (15.0–60.0) against diagnosed PCC (coding), although it is less effective in younger age groups and not effective at 18 months since the last vaccination [40]. Many pharmaceutical candidates are undergoing trials to establish if there are effective therapeutic interventions [41].

### Strengths

Existing literature recommends that longitudinal cohort studies include control groups, record symptoms at consistent follow-up intervals, and consider pre-existing medical conditions.[42] A strength of this study is that it attempts to address these limitations by incorporating these factors in the design. Whilst it is no longer feasible to prospectively compare post-acute sequelae in children and adolescents who have never had COVID-19, this design allows a comparison with another common viral infection, influenza. Additionally, the study uses PedsQL outcome measurement tools, part of a core outcome set developed through a Delphi consensus process by an international and multidisciplinary group [43].

### Limitations

The primary limitation is decreasing COVID-19 notifications, partly due to reduced testing and notifications to SA Health. The current recommendation for COVID-19 testing in South Australia is that rapid antigen testing (RAT) no longer needs to be reported to SA Health, SA Government. However, there is still an option to report a positive RAT test if someone wishes to do so. Earlier in the pandemic, individuals in close contact with a COVID-19 case or any cold or flu-like symptoms were tested and notified to SA Health if COVID-19 was detected. However, this is no longer the case, and changes to testing may result in selection bias due to COVID-19 cases being tested because they are at higher risk of

complications from COVID-19 or influenza or are more acutely unwell at presentation. Additionally, people with ongoing symptoms may be more inclined to consent to participate in the study.

The study design, which recruits participants approximately 1 month post-COVID-19 or influenza infection, attempts to mitigate some of this bias, as the 3-month questionnaire is the first time point where they can be classified as having PCC or PAS following influenza. The timing of the 1-month questionnaire also attempts to reduce recall bias by asking about initial symptoms closer to their acute illness.

Despite the study's limitations, the findings will contribute to filling knowledge gaps with respect to the clinical features, natural history and risk factors for PCC and PAS following influenza in children and adolescents, offering contemporary evidence generalizable to infections during Omicron circulation.

## Supporting information

**S1 Appendix. Definitions used for post-COVID-19 condition (PCC) and post-acute sequelae (PAS) following influenza, adapted from the World Health Organization.**
(DOCX)

**S2 Appendix. SPIRIT Checklist.**
(DOCX)

**S3 Appendix. Study protocol.**
(PDF)

## Author contributions

**Conceptualization:** Mark McMillan, Thomas R Sullivan, Philip N Britton, Brendan McMullan, Helen S Marshall.

**Funding acquisition:** Mark McMillan.

**Methodology:** Mark McMillan, Rebecca Beazley, Nan Vasilunas, Thomas R Sullivan, Tess Edmond, Ailish Battersby, Philip N Britton, Brendan McMullan, Jon Jureidini, Helen S Marshall.

**Project administration:** Mark McMillan, Rebecca Beazley.

**Resources:** Mark McMillan, Rebecca Beazley, Sarah Del Fante.

**Software:** Mark McMillan.

**Supervision:** Mark McMillan, Rebecca Beazley, Helen S Marshall.

**Writing – original draft:** Mark McMillan.

**Writing – review & editing:** Rebecca Beazley, Nan Vasilunas, Thomas R Sullivan, Tess Edmond, Ailish Battersby, Philip N Britton, Brendan McMullan, Jon Jureidini, Sarah Del Fante, Helen S Marshall.

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
