## [Decision Letter · Decision Letter 0]

1 Sep 2025

Dear Dr. McMillan,

Thank you for submitting your manuscript to PLOS ONE. After careful consideration, we feel that it has merit but does not fully meet PLOS ONE’s publication criteria as it currently stands. Therefore, we invite you to submit a revised version of the manuscript that addresses the points raised during the review process.

We look forward to receiving your revised manuscript.

Kind regards,

Katherine Demi Kokkinias, Ph.D.

Staff Editor

PLOS ONE

Journal Requirements:

3. Please include captions for your Supporting Information files at the end of your manuscript, and update any in-text citations to match accordingly. Please see our Supporting Information guidelines for more information: http://journals.plos.org/plosone/s/supporting-information .

4. We note that the original protocol file you uploaded contains a confidentiality notice indicating that the protocol may not be shared publicly or be published. Please note, however, that the PLOS Editorial Policy requires that the original protocol be published alongside your manuscript in the event of acceptance. Please note that should your paper be accepted, all content including the protocol will be published under the Creative Commons Attribution (CC BY) 4.0 license, which means that it will be freely available online, and any third party is permitted to access, download, copy, distribute, and use these materials in any way, even commercially, with proper attribution.

Therefore, we ask that you please seek permission from the study sponsor or body imposing the restriction on sharing this document to publish this protocol under CC BY 4.0 if your work is accepted. We kindly ask that you upload a formal statement signed by an institutional representative clarifying whether you will be able to comply with this policy. Additionally, please upload a clean copy of the protocol with the confidentiality notice (and any copyrighted institutional logos or signatures) removed.

Additional Editor Comments:

Please note that we have only been able to secure a single reviewer to assess your manuscript. We are issuing a decision on your manuscript at this point to prevent further delays in the evaluation of your manuscript. Please be aware that the editor who handles your revised manuscript might find it necessary to invite additional reviewers to assess this work once the revised manuscript is submitted. However, we will aim to proceed on the basis of this single review if possible.

The reviewer has raised a number of major concerns. A reviewer had questions and suggestions regarding the methodology and statistics that will be used in the study. Specifically, the reviewer required a more detailed description of the statistical parameters of the study including definitions of terms, how prediction models will be developed, and how missing data will be handled.

Could you please carefully revise the manuscript to address all comments raised?

Reviewers' comments:

Reviewer's Responses to Questions

**Comments to the Author**

1. Does the manuscript provide a valid rationale for the proposed study, with clearly identified and justified research questions?

Reviewer #1: Yes

2. Is the protocol technically sound and planned in a manner that will lead to a meaningful outcome and allow testing the stated hypotheses?

Reviewer #1: Yes

3. Is the methodology feasible and described in sufficient detail to allow the work to be replicable?

Reviewer #1: No

4. Have the authors described where all data underlying the findings will be made available when the study is complete?

Reviewer #1: No

5. Is the manuscript presented in an intelligible fashion and written in standard English?

Reviewer #1: Yes

You may also provide optional suggestions and comments to authors that they might find helpful in planning their study.

Reviewer #1: This protocol describes a study that aims to compare the incidence of PCC and PAS in children.

Many details are missing from the statistical analysis section and elsewhere. See comments below.

1. Please add the PAS and PCC definitions to the document.

2.The statistical analysis section does not match the study aims and objectives. Also, there are many secondary objectives (e.g. estimating the relative risk of developing PCC vs PAS) that are not addressed in the statistical analysis section. Please add details about these.

3. How will person time be calculated? How will subjects that drop out be dealt with.

2. "The incidence of PCC and PAS will be estimated" - at which time point?

3. Will a Poisson distribution be used?

4. Will the incidences of PCC and PAS be compared? How? Should these be adjusted for confounders? Which ones?

5. Will a logistic regression model be used to develop the predication model? At what time point? How will subjects that drop out be treated? Which covariates will be considered? How will continuous covariates be parametrized? Please specify which measures will be used to evaluate the prediction model.

Given the small number of cases expected of PCC (12-72) this means that the prediction model could have 1-5 degrees of freedom. So very few predictors. Please discuss as a limitation.

**Do you want your identity to be public for this peer review?** For information about this choice, including consent withdrawal, please see our Privacy Policy

Reviewer #1: No

---

## [Author Response · Author response to Decision Letter 1]

5 Dec 2025

Thank you for the opportunity to revise our manuscript and for the thoughtful feedback provided by the reviewer. We greatly appreciate the reviewer’s careful assessment and constructive comments, which have helped strengthen the protocol.

Reviewer #1: This protocol describes a study that aims to compare the incidence of PCC and PAS in children.

Many details are missing from the statistical analysis section and elsewhere. See comments below.

1. Please add the PAS and PCC definitions to the document.

Definitions for Post-Acute Sequelae (PAS) and Post-COVID Condition (PCC) have been added to the Supplementary Appendix and are referenced in the main manuscript at line 171

2.The statistical analysis section does not match the study aims and objectives. Also, there are many secondary objectives (e.g. estimating the relative risk of developing PCC vs PAS) that are not addressed in the statistical analysis section. Please add details about these.

The statistical analysis section has been expanded and aligned with the study aims. We now specify the estimation of cumulative incidence at 3 months post-infection and proportions with ongoing symptoms at 6 and 12 months. The secondary objectives have been reworded and condensed for clarity. The revised objectives and detailed analytical methods are provided below:

Line 90:

Primary objective

1. Estimate the cumulative incidence of PCC in children and adolescents with COVID-19 infection at 3 months, and the proportion with ongoing symptoms at 6 and 12 months post-COVID-19 infection.

Secondary objectives

1) Estimate the cumulative incidence of PAS in children and adolescents with influenza infection at 3 months, and the proportion with ongoing symptoms at 6 and 12 months.

2) Estimate the relative risk of developing PCC following COVID-19 compared to PAS following influenza at 3 months post-infection.

3) Develop a prediction model for the development of PCC in children and adolescents with COVID-19.

4) Estimate the difference in the quality of life of children and adolescents (aged 2-18) at 3, 6, and 12 months post-infection in the following groups:

i) PCC compared to those without PCC in COVID-19-positive cases.

ii) PAS compared to those without PAS in influenza-positive cases.

iii) PCC compared to those with PAS

5) Estimate the difference in fatigue of children and adolescents (aged 2-18) at 3, 6, and 12-months post-infection in the following groups:

i) PCC compared to those without PCC in COVID-19-positive cases.

ii) PAS compared to those without PAS in influenza-positive cases.

iii) PCC compared to those with PAS

6) Estimate the difference in school/childcare absenteeism at 3, 6, and 12 months following infection in the following groups:

i) PCC compared to those without PCC in COVID-19-positive cases.

ii) PAS compared to those without PAS in influenza-positive cases.

iii) PCC compared to those with PAS

We have included further details about the planned statistical methods and linked these to the objectives above. The new wording can be found starting line 199:

Analyses will follow a pre-specified statistical analysis plan. The cumulative incidence of PCC following COVID-19 (primary objective) and PAS following influenza (secondary objective 1) will be estimated at 3 months post-infection, with 95% confidence intervals. The proportion of individuals with ongoing symptoms will be reported descriptively (n, %) at 6 and 12 months. The relative risk of developing PCC following COVID-19 versus developing PAS following influenza (objective 2) will be estimated using a log-binomial model. A multivariable logistic model will be developed for predicting PCC at 3 months (objective 3), incorporating pre-specified predictors such as age (treated as continuous), sex, pre-existing conditions, and socioeconomic status. Model performance will be evaluated using the area under the receiver operating characteristic curve, with internal validation by bootstrap resampling in line with TRIPOD recommendations.[37] The relatively small expected number of PCC events may limit the number of predictors that can be incorporated into the final model and hence its predictive accuracy. Linear and negative binomial mixed-effects models will assess differences between comparator groups over time in continuous (PedsQL quality of life and fatigue scores, objectives 4 and 5) and count (days absent, objective 6) outcomes, respectively. Log binomial models will compare the risk of developing PAS according to influenza vaccine uptake (objective 7) and the risk of PCC by COVID-19 reinfection (objective 8). The duration and clinical features of PCC and PAS will be described using summary statistics, with summaries at 3, 6 and 12 months.

3. How will person time be calculated? How will subjects that drop out be dealt with.

We have clarified in the objectives that we are looking at the cumulative incidence at 3 months, and then the proportion of cases with ongoing symptoms at 6 and 12 months. Hence, person-time will not be relevant to the analysis. In participants that develop PCC (as defined at 3 months), drop-out at 6 or 12 months will entail missing data for determining whether symptoms were ongoing at these time-points.

The following wording has been added to line 226 to clarify the approach:

Handling of follow-up and missing data.

Outcomes will be summarised at the planned 3, 6 and 12-month assessments. Participants contribute data to each analysis for which follow-up information is available. For the primary objective, participants with PCC at 3 months who discontinue or miss assessments at 6 or 12 months will be treated as missing for whether symptoms were ongoing at these time-points. Multiple imputation may be considered for the prediction model, depending on the amount and reasons for missing data at 3 months when determining PCC.

4. "The incidence of PCC and PAS will be estimated" - at which time point?

We have clarified this in the objectives that cumulative incidence will be estimated at 3 months

5. Will a Poisson distribution be used?

We have clarified in the objectives that cumulative incidence will be estimated at 3 months and a Poisson distribution will not be required.

6. Will the incidences of PCC and PAS be compared? How? Should these be adjusted for confounders? Which ones?

The relative risk of developing PCC following COVID-19 versus developing PAS following influenza will be estimated using a log-binomial model. Adjustment for confounders is not planned, as PCC and PAS arise from distinct exposure groups (COVID-19 vs influenza) and the outcome itself (PCC or PAS) is defined differently for these two groups.

7. Will a logistic regression model be used to develop the predication model? At what time point? How will subjects that drop out be treated? Which covariates will be considered? How will continuous covariates be parametrized? Please specify which measures will be used to evaluate the prediction model.

Given the small number of cases expected of PCC (12-72) this means that the prediction model could have 1-5 degrees of freedom. So very few predictors. Please discuss as a limitation.

The following wording has been added to the manuscript line 205:

A multivariable logistic model will be developed for predicting PCC at 3 months (objective 3), incorporating pre-specified predictors such as age (treated as continuous), sex, pre-existing conditions, and socioeconomic status. Model performance will be evaluated using the area under the receiver operating characteristic curve, with internal validation by bootstrap resampling in line with TRIPOD recommendations.[37] The relatively small expected number of PCC events may limit the number of predictors that can be incorporated into the final model and hence its predictive accuracy.

---

## [Decision Letter · Decision Letter 1]

9 Jan 2026

Children and Adolescents: Respiratory Infection and Long-term Effects Longitudinal Study (CARE Study): Study Protocol

PONE-D-25-34138R1

Dear Dr. McMillan,

We’re pleased to inform you that your manuscript has been judged scientifically suitable for publication and will be formally accepted for publication once it meets all outstanding technical requirements.

Kind regards,

Shibajee Debbarma, M.D.

Academic Editor

PLOS One

Additional Editor Comments (optional):

Reviewers' comments:

Reviewer's Responses to Questions

**Comments to the Author**

1. Does the manuscript provide a valid rationale for the proposed study, with clearly identified and justified research questions?

Reviewer #1: Yes

2. Is the protocol technically sound and planned in a manner that will lead to a meaningful outcome and allow testing the stated hypotheses?

Reviewer #1: Yes

3. Is the methodology feasible and described in sufficient detail to allow the work to be replicable?

Reviewer #1: Yes

4. Have the authors described where all data underlying the findings will be made available when the study is complete?

Reviewer #1: No

5. Is the manuscript presented in an intelligible fashion and written in standard English?

Reviewer #1: Yes

You may also provide optional suggestions and comments to authors that they might find helpful in planning their study.

Reviewer #1: The authors have done an excellent job addressing most of my points and the revision is very good.

However, despite estimating the relative risk of a negative outcome for COVID vs. flu, the analysis will not be adjusted for confounders. It seems likely to me that there may be differences in the patient populations that are related both to the risk of getting COVID or flu, and to the risk of experiencing the outcome. An adjusted model seems necessary.

**Do you want your identity to be public for this peer review?** For information about this choice, including consent withdrawal, please see our Privacy Policy

Reviewer #1: No

---

## [Editor Report · Acceptance letter]

PONE-D-25-34138R1

PLOS One

Dear Dr. McMillan,

I'm pleased to inform you that your manuscript has been deemed suitable for publication in PLOS One. Congratulations! Your manuscript is now being handed over to our production team.

Kind regards,

on behalf of

Dr. Shibajee Debbarma

Academic Editor

PLOS One